# Evolution in caesarean section practices in North Kivu: Impact of caregiver training

**Michel Dikete Ekanga**[1]*, **Prudence Mitangala**[2], **Yves Coppieters**[3], **Christine Kirkpatrick**[1], **Richard Kabuyanga Kabuseba**[4], **Philippe Simon**[1], **Yvon Englert**[1], **Judith Racape**[3,5], **Wei-Hong Zang**[3,6]

1 Obstetric Gynecology Department, Erasmus Hospital, Free University of Brussels, University Clinics of Brussels, Bruxelles, Belgium, 2 North Kivu Provincial Health Division, Goma, DR Congo, 3 School of Public Health, Center for Research in Epidemiology, Biostatistics and Clinical Research, Free University of Brussels, Brussels, Belgium, 4 North Kivu Provincial Hospital, Goma, DR Congo, 5 Chair in Health and Precarity, Faculty of Medecine (ULB), Universite Libre de Bruxelles, Brussels, Belgium, 6 Department of Public Health and Primary Care, Faculty of Medicine and Health Sciences, International Center for Reproductive Health, Ghent University, Ghent, Belgium

* Michel.Dikete.Ekanga@erasme.ulb.ac.be

## Abstract

### Introduction

A caesarean section is a major obstetric procedure that can save the life of mother and child. Its purpose is to protect the mother's health from the complications of childbirth and to protect the baby's health. In sub-Saharan Africa (SSA), there are major inequalities in access to caesarean sections and significant variations in practices to determine the indications for the procedure. Periodic analyses of maternal deaths have shown that more than half of maternal and new born deaths are due to suboptimal care and are therefore potentially preventable. The objective of our study is to assess the impact of health staff training under the PADISS project (to support the health system's integrated development) on the quality of CS procedures in North Kivu, by comparing two periods.

### Material and methods

The populations compared were recruited from the referral hospitals in North Kivu, DRC (Democratic Republic of Congo). The first (group 1) was made up of patient files studied retrospectively for the period from 01/11/2013 to 01/01/2016. The second group (group 2), studied prospectively, comprised patient files from June 2019 to January 2020. Obstetric, maternal and foetal data were compared. Statistical analyses were performed using STATA/IC 15.0 for Windows. Univariate and multiple logistic regression was performed to determine which characteristics are associated with maternal and perinatal morbidity and mortality. A p value < 0.05 was considered statistically significant.

### Results

CS frequency was approximately 17% in both study periods. We observed a CS frequency of about 34% at North Kivu provincial hospital for the two populations studied. The main indications for CS were dystocia, foetal distress and scarred uterus for both populations. In the

**Data Availability Statement:** The minimal data set is contained in the paper and Supporting Information files. For the raw data used in this

paper, please send a written request to the Obstetric Gynecology department. Belgium, Erasmus Hospital, Lennik Road 808,1070 Anderlecht at the following email address: Michel. Dikete.Ekanga@erasme.ulb.ac.be.

**Funding:** The author(s) received no specific funding for this work.

**Competing interests:** The authors have declared that no competing interests exist.

population studied prospectively, after the implementation of health staff training, there were fewer incidence rate of dystocia, foetal distress and neonatal death, a more complete patient record, shorter hospital stay, and fewer blood transfusions but more incidence rate of scarred uterus, post-operative complications and low birth weight. Intervention had no statistically significant impact on low birth weight (OR = 1.9, p = 0.13), on neonatal mortality (OR = 0.69, p = 0.21).

## Conclusion

Our study shows a decrease in neonatal deaths, dystocia and foetal distress, but an increase in post-operative complications, maternal deaths and cases of scarred uterus and low birth weight. However, multiple logistic regression did no support the conclusion.

## Introduction

A caesarean section is a major obstetric procedure that can save the life of mother and child [1]. Its purpose is to protect the mother's health from the complications of labour (uterine rupture, fistulae, perineal problems and incontinence, postpartum hemorrhage, placenta praevia) or to protect the baby's health (stillbirth, asphyxia, neurological damage). In sub-Saharan Africa (SSA), there are major inequalities in access to caesarean sections and significant variations in practices to determine the indications for the procedure. On the one hand, financial, geographic and cultural barriers deprive women of a procedure that could save their lives. On the other hand, the growing use of caesarean section with no medical indication, in a context of poor quality of care, causes iatrogenic and preventable excess morbidity and mortality [2, 3].Worldwide, approximately 22.9 million caesarean sections (CS) are performed every year, primarily to save the life of the mother and/or newborn infant [1]. However, with nearly 2 deaths in 1000 live births, the global post-CS neonatal mortality rate is three times higher than the vaginal delivery mortality rate [4]. Nearly 300,000 women die every year as a result of a pregnancy, a caesarean or vaginal delivery [5]. Of these maternal deaths, 99% occur in developing countries [5]. In sub-Saharan African (SSA), 8.8% of deliveries are by CS [6], which is in line with the 10 to 15% recommended by the World Health Organization (WHO) [7]. However, intrapartum mortality in SSA accounts for 73% of neonatal deaths worldwide [8]. In the Democratic Republic of Congo (DRC), the latest demographic and health surveys show that the CS rate rose from 4% in 2007 [9] to 5% in 2013 [9]. A study conducted in Goma (North Kivu) between 2013 and 2016 in four referral hospitals reported a caesarean rate of 16%, that the majority of CS were performed as an emergency and under loco regional anaesthesia, and that the perinatal risk was higher when the CS was performed as an emergency and under loco regional anaesthesia [10]. Several risk factors which increase maternal and perinatal morbidity and mortality in the intra- and post-CS period have been identified, in particular the shortage of qualified health staff, lack of health staff training, inadequate medical infrastructure, difficulties in performing the CS in due time when women need one, as well as the high cost of the procedure in relation to the population's income [11]. Periodic analyses of maternal deaths in the United Kingdom have shown that more than half of maternal and neonatal deaths were due to suboptimal care and that they were therefore potentially preventable [12]. The authors highlight not only the inability of certain health staff to identify and treat obstetric emergencies, and gaps in technical skills, but also significant problems as regards interpersonal and interdisciplinary communication. In the United States, the Institute of Medicine has published

two reports that highlight the importance of improving the quality of healthcare [13, 14] and recommend establishing training programmes for individuals and teams that incorporate methods such as simulation, which has proven effective in other fields of medicine. A programme carried out recently in Tanzania, showed the potential of simulation for decreasing the occurrence of obstetric complications in countries with few material resources, improving particularly the management of the third stage of labour [15]. The objective of our study is to assess the impact of the health staff training provided under the PADISS project (to support the health system's integrated development) on the quality of CS procedures in North Kivu province.

## Material and methods

### Population

The first, baseline population was recruited from referral hospitals in the city of Goma from the period 01/11/2013 to 01/01/2016 (group 1). It is a retrospective, descriptive and analytical study concerning all CS performed across all deliveries that occurred during the period under study [10]. Twin pregnancies were excluded from the study. The medical staffs in each hospital were capable of performing a CS. A data collection form was designed. Data collection was carried out by a team of investigators made up of doctors and midwives in these maternity units. The sources of information were the delivery record, partograph, surgical reports and neonatal records. The sociodemographic parameters (maternal age, marital status, level of education, ethnicity, occupation, primary residence, weight, height), medical and surgical history, obstetric environment (antenatal monitoring), and maternal and perinatal morbidity and mortality (complications and outcome) were analyzed.

The data collection was anonymous for the retrospective study after the agreement of the provincial authority. For the prospective study, consent was verbal, free and informed after explaining the objectives of this research.

### Intervention

After analysis of our retrospective study, cesarean section was not a factor in reducing maternal and perinatal morbidity and mortality, hence the benefit of improving working conditions at the level of referral centers, transfer conditions, basic infrastructure and caregiver training.

Following these findings, specific training was given to the local health staff. The training was based around theory and clinical activities that were carried out during our various visits in order to increase the skills of the gynaecologists, obstetricians and midwives in the different health zones in North Kivu. This training was delivered by the team of gynaecologists, obstetricians, paediatricians, anaesthetists, hygienists, midwives and theatre nurses at Erasme hospital in Brussels as part of the PADISS project. For the theory activities, several seminars were held on the pedagogical elements with a continuing education frame of reference, on the organisation of clinical audits, the preparation of treatment protocols with all the gynaecologists and members of the provincial health department (DPS), the preparation of different themes for the continuing education, such as gestational hypertension, gestational diabetes, the management of antenatal and post-partum hemorrhage, caesarean section and its alternatives, echography in gynaecology and obstetrics, antenatal consultations, monitoring during labour, the use of obstetric manoeuvres, hospital hygiene, the maintenance of medical files, and neonatal care. The gynaecologists, obstetricians, midwives and maternity-paediatric nurses from these various health zones attended these non-certificate courses on a daily basis.

For the clinical activities, the objective was to provide technical support to medical and midwifery trainers, establish the pedagogical method for the training linking the frame of reference and the different activities, and carry out individual skills assessments. The clinical activities were based around ward rounds with trainee doctors and by the provincial trainer, participation in gynaecological and obstetrical consultations, the approach to paraclinical tests taking into account the situation on the ground, assistance in the operating theatre and delivery room, as well as teaching obstetric maneuvers to all health staff using simulations on a manikin. A second population (group 2) was therefore selected prospectively in order to assess the benefit of the training provided. The training started from February 2016 until March 2020. The various teams of gynecologists, pediatricians, anesthesiologists, health doctors, midwives, nurses in the operating room had carried out several missions in the field. Each mission was for an average duration of 10 days, 8 hours of theoretical and practical training per day.

The second group was recruited from the same hospitals in Goma for the period from June 2019 to January 2020 (group 2). This was a prospective, analytical, cross-disciplinary study concerning all CSs performed across all deliveries (2094) that occurred during the period under study. Twin pregnancies were excluded from the study. All the women who had a CS in the hospitals were included exhaustively in the study. Data collection was carried out by a team of investigators made up of doctors and midwives in these maternity units. The sources of information were the delivery record, partograph, surgical reports and neonatal records. The sociodemographic parameters (maternal age, marital status, level of education, ethnicity, occupation, primary residence, weight, height), medical and surgical history, obstetric environment (antenatal visits), and maternal and perinatal morbidity and mortality (complications and outcome) were analyzed and compared with group 1.

## Statistical analysis

The findings are reported as a percentage for the categorical variables, as average and standard deviation (SD) or median and interquartile range [25%-75%] for the quantitative variables depending on their respective distribution, Gaussian or otherwise. The categorical variables were compared between the two groups using the Chi$^2$ Pearson test, or Fisher's exact test for a small sample. The quantitative variables were compared using the Mann Whitney test depending on their respective distribution, Gaussian or otherwise.

Univariate and multiple logistic regression was used to study the association between our variables (socio-demographic, medical and obstetric) and three other variables (low birth weight, neonatal mortality and post-operative complications). The variables with a high percentage of missing values were not included in the univariate analysis. The variables included in our multiple logistic regression model were selected according to the statistical association ($p \leq 0.05$) with the result of the univariate analysis, and to the total number of cases. The odds ratios and their 95% confidence intervals were calculated using each variable coefficient (and standard errors) in the model. The significance of each coefficient was tested using the Wald test. The Hosmer and Lemeshow test was used to verify the model's goodness of fit. A value of $p < 0.05$ was considered statistically significant. Statistical analyses were performed using STATA/IC 16.0 for Windows.

Ethic statement: The provincial health division of North-Kivu in the democratic Republic of Congo does not have an ethics commitee. Authorization to conduct this research was obtained by the same provincial authority that had waived informed consent for the retrospective study. The data in this retrospective study was anonymised.

For the prospective study, patients had given their informed verbal consent for data from their medical records to be used for research.

## Results

### CS frequency and sociodemographic characteristics

The CS frequency in the study prior to our intervention (group 1) was 16.2%, and after our intervention (group 2), the CS frequency was 17%. This difference is not statistically significant.

The highest frequency at approximately 34% was seen at the North Kivu provincial hospital in both groups. Table 1 shows the socio-demographic characteristics of the women whose files were included in the study. Overall, the women who had undergone a CS in the study prior to our intervention (group 1) were mostly married, housewives, who had been to secondary school, were of Nande ethnicity, and living in the vicinity of the medical facility that performed the CS (Table 1).Our study showed more single women, with a lower level of education, and

**Table 1. Socio-demographic data.**

|  | No Intervention (n = 694) | Intervention (n = 356) | p-value |
|---|---|---|---|
| **Woman's age (years)*** |  |  | 0.81 |
|  | 26.8 (6.2) | 26.7 (5.7) |  |
| **Marital status** |  |  | 0.001 |
| Married | 604 (89.1) | 281 (81.5) |  |
| Single | 74 (10.9) | 64 (18.5) |  |
| **Woman's level of education** |  |  | < 0.0001 |
| None | 82 (12.4) | 100 (28.7) |  |
| Primary | 62 (9.4) | 93 (26.7) |  |
| Secondary | 301 (45.4) | 125 (35.9) |  |
| Higher | 87 (13.1) | 29 (8.3) |  |
| Other | 131 (19.8) | 1 (0.3) |  |
| **Ethnicity#** |  |  | < 0.0001 |
| Nande | 165 (38.6) | 162 (46.4) |  |
| Hunde | 59 (13.8) | 43 (12.3) |  |
| Havu | 13 (3.0) | 15 (4.3) |  |
| Shi | 47 (11.0) | 32 (9.2) |  |
| Hutu | 33 (7.7) | 52 (14.9) |  |
| Other | 110 (25.8) | 45 (12.9) |  |
| **Woman's occupation** |  |  | < 0.0001 |
| Housewife | 526 (77.2) | 193 (54.7) |  |
| Civil servant | 20 (2.9) | 11 (3.1) |  |
| Student | 24 (3.5) | 4 (1.1) |  |
| Liberal profession | 16 (2.4) | 17 (4.8) |  |
| Farmer | 9 (1.3) | 91 (25.8) |  |
| Other | 86 (12.6) | 37 (10.5) |  |
| **Primaryresidence** |  |  | < 0.0001 |
| Urban area | 617 (92.8) | 220 (61.8) |  |
| Rural area | 48 (7.2) | 136 (38.2) |  |
| **Hospital in the health zone of the woman's village of origin** |  |  | < 0.0001 |
| Yes | 405 (62.8) | 160 (48.3) |  |
| No | 240 (37.2) | 171 (51.7) |  |

*mean (SD)

#38.5% missing

farmers living in rural areas far from the hospital in the health zone of their village of origin in group 2 than in group 1.

## Medical data

In group 2, the height and weight of the women was complete and properly recorded in the files and this made it possible to calculate the BMI (body mass index) more accurately. In group 2, primigravidas underwent fewer CSs, and the CS was mostly performed on women who had a history of two or more CS. There was a higher number of women who had been to four antenatal visits or more in group 2 (Table 2). In group 2, urine tests during the antenatal

**Table 2. Medical data.**

|  |  | No Intervention (n = 694) | Intervention (n = 356) | p-value |
|---|---|---|---|---|
| **BMI (kg/m$^2$)**$^*$ |  |  |  | < 0.0001 |
|  |  | 28.9 (4.7) | 27.1 (4.0) |  |
| **History of hypertension** |  |  |  | < 0.0001 |
|  | No | 175 (87.1) | 287 (96.3) |  |
|  | Yes | 26 (17.9) | 11 (3.7) |  |
| **Family history of hypertension** |  |  |  | < 0.0001 |
|  | No | 113 (75.8) | 270 (92.1) |  |
|  | Yes | 36 (24.2) | 23 (7.9) |  |
| **History of diabetes** |  |  |  | 0.28 |
|  | No | 196 (97.5) | 295 (99) |  |
|  | Yes | 5 (2.5) | 3 (1) |  |
| **Family history of diabetes** |  |  |  | 0.1 |
|  | No | 130 (87.2) | 270 (92.2) |  |
|  | Yes | 19 (12.8) | 23 (7.8) |  |
| **Parity** |  |  |  | < 0.0001 |
|  | 0 | 210 (30.6) | 65 (18.2) |  |
|  | 1–3 | 319 (46.4) | 195 (54.8) |  |
|  | $\geq 4$ | 158 (23) | 96 (27) |  |
| **Abortion** |  |  |  | 0.07 |
|  | 0 | 519 (78.9) | 286 (81) |  |
|  | 1 | 91 (12.8) | 54 (15.3) |  |
|  | $\geq 2$ | 48 (7.3) | 13 (3.7) |  |
| **Number of previous caesareans** |  |  |  | < 0.0001 |
|  | 0 | 304 (52.4) | 104 (32) |  |
|  | 1 | 154 (26.6) | 88 (27.1) |  |
|  | $\geq 2$ | 122 (21) | 133 (40.9) |  |
| **No. of antenatal visits during current pregnancy** |  |  |  | 0.02 |
|  | 0 | 16 (2.7) | 5 (1.5) |  |
|  | 1 | 26 (4.4) | 13 (4.0) |  |
|  | 2 | 69 (11.6) | 22 (6.8) |  |
|  | 3 | 188 (31.7) | 89 (27.6) |  |
|  | $\geq 4$ | 294 (49.6) | 193 (59.9) |  |
| **Albuminuria** |  |  |  | < 0.0001 |
|  | Yes | 82 (13.6) | 8 (2.3) |  |
|  | No | 202 (33.4) | 211 (60.1) |  |
|  | Not investigated | 320 (53) | 132 (37.6) |  |
| **Sugar in the urine** |  |  |  | < 0.0001 |

*(Continued)*

**Table 2.** (Continued)

| | No Intervention (n = 694) | Intervention (n = 356) | p-value |
|---|---|---|---|
| Yes | 39 (6.5) | 1 (0.3) | |
| No | 181 (30.3) | 211 (59.9) | |
| Not investigated | 378 (63.2) | 140 (39.8) | |
| **Urine infection** | | | < 0.0001 |
| Yes | 79 (29.4) | 51 (14.5) | |
| No | 249 (40.9) | 203 (57.7) | |
| Not investigated | 180 (29.6) | 98 (27.8) | |
| **Presence of malaria parasite** | | | < 0.0001 |
| Yes | 68 (11.3) | 18 (5.1) | |
| No | 323 (23.6) | 139 (39.6) | |
| Not investigated | 211 (35.1) | 194 (55.3) | |
| **HIV** | | | 0.89 |
| Yes | 7 (1.1) | 4 (1.2) | |
| No | 440 (72.5) | 255 (73.9) | |
| Not investigated | 160 (26.4) | 86 (24.9) | |
| **Rubella** | | | 0.003 |
| Yes | 1 (0.2) | 1 (0.3) | |
| No | 35 (5.9) | 40 (11.5) | |
| Not investigated | 561 (94) | 307 (88.2) | |
| **Toxoplasmosis** | | | < 0.0001 |
| Positive | 2 (0.3) | 0 | |
| Negative | 12 (2.0) | 24 (6.9) | |
| Not investigated | 590 (97.7) | 323 (93.1) | |
| **CMV** | | | 0.002 |
| Positive | 0 | 1 (0.3) | |
| Negative | 12 (2.0) | 20 (5.8) | |
| Not investigated | 591 (98) | 324 (93.9) | |

visits highlighted less albumin, less sugar and fewer urinary tract infections in relation to group 1 (Table 2). Blood tests at the antenatal visits in group 2 showed few cases of malaria, but this was not investigated in the majority of patients. Toxoplasmosis, cytomegalovirus and rubella status was not widely investigated in either group. The majority of women in both study groups were HIV negative (Table 2).

## Obstetric characteristics

The majority of these CSs were performed as emergencies and for cephalic presentation of the foetus, and loco regional anaesthesia was the most used in both groups. However, many more patients had been referred before onset of labour and during labour in group 2 than in group 1. There were more spontaneous and fewer induced labours, and more CSs performed before onset of labour in group 2 than in group 1 (Table 3). The main indications for CS in both groups were dystocia, scarred uterus and foetal distress. In group 2, there were fewer cases of dystocia and foetal distress, but more cases of scarred uterus, a more comprehensive operating protocol, more transverse incisions, fewer blood transfusions, and the CS was performed more often by a gynaecologist than in group 1 (Table 3).

**Table 3. Obstetric data.**

| | No Intervention (n = 694) | Intervention (n = 356) | p-value |
|---|---|---|---|
| **Type of admission** | | | 0.9 |
| Normal prior to onset of labour | 260 (38.3) | 133 (37.9) | |
| Emergency during labour | 419 (61.7) | 218 (62.1) | |
| **Mode of admission** | | | < 0.0001 |
| Referral prior to onset of labour | 5 (0.7) | 149 (42.1) | |
| Referral during labour | 131 (19.3) | 186 (52.5) | |
| Personal decision prior to onset of labour | 139 (20.5) | 7 (2.0) | |
| Personal decision during labour | 403 (59.4) | 12 (3.4) | |
| **If transfer, facility situation** | n = 144 | n = 332 | 0.06 |
| Facility in the hospital's health zone | 87 (60.4) | 169 (50.9) | |
| Facility outside the hospital's health zone | 57 (39.6) | 163 (49.1) | |
| **Fundal height in centimetres before caesarean** | | | 0.01 |
| | 32.5 (3.2) | 33.1 (3.6) | |
| **Presentation of the foetus** | | | 0.01 |
| Cephalic | 611 (90.3) | 314 (89.2) | |
| Breech | 53 (7.8) | 20 (5.7) | |
| Transverse lie | 8 (1.2) | 15 (4.3) | |
| Other | 5 (0.7) | 3 (0.9) | |
| **Onset of labour** | | | < 0.0001 |
| Spontaneous | 517 (76.8) | 282 (80.8) | |
| Induced | 124 (18.4) | 11 (3.1) | |
| Caesarean before labour | 32 (4.8) | 56 (16.1) | |
| **Type of primary anaesthesia** | | | 0.002 |
| loco-regional | 429 (62.8) | 255 (72.4) | |
| general | 254 (37.2) | 97 (27.6) | |
| **Type of secondary anaesthesia** | | | 0.004 |
| No | 625 (97) | 319 (92.5) | |
| General | 9 (1.4) | 14 (4.1) | |
| Loco-regional | 10 (1.6) | 12 (3.5) | |
| **Pre-operative indication for Caesarean 1** | | | |
| **Pelvic anomaly** | | | |
| No | 568 (82.4) | 316 (91.9) | < 0.0001 |
| Yes | 121 (17.6) | 28 (8.1) | |
| **Scarred uterus** | | | |
| No | 640 (92.9) | 316 (91.9) | 0.55 |
| Yes | 49 (7.1) | 28 (8.1) | |
| **Foetal distress** | | | |
| No | 597 (86.7) | 309 (89.8) | 0.14 |
| Yes | 92 (13.3) | 35 (10.2) | |
| **Placenta praevia** | | | |
| No | 663 (96.2) | 334 (97.1) | 0.47 |
| Yes | 26 (3.8) | 10 (2.9) | |
| **Other** | | | |
| No | 433 (62.8) | 235 (68.3) | 0.08 |
| Yes | 256 (37.2) | 109 (31.7) | |
| **Post-operative indication for Caesarean 1** | | | |
| **Pelvic anomaly** | | | |

*(Continued)*

**Table 3.** (Continued)

| | No Intervention (n = 694) | Intervention (n = 356) | p-value |
|---|---|---|---|
| No | 564 (82.9) | 280 (89.7) | 0.005 |
| Yes | 116 (17.1) | 32 (10.3) | |
| **Scarred uterus** | | | |
| No | 532 (78.2) | 201 (64.6) | < 0.0001 |
| Yes | 148 (21.8) | 110 (35.4) | |
| **Functional dystocia** | | | |
| No | 636 (93.5) | 287 (92.3) | 0.47 |
| Yes | 44 (6.5) | 24 (7.7) | |
| **Foetal distress** | | | |
| No | 582 (85.6) | 277 (89.4) | 0.11 |
| Yes | 98 (14.4) | 33 (10.6) | |
| **Placenta praevia** | | | |
| No | 655 (96.3) | 301 (97.1) | 0.54 |
| Yes | 25 (3.7) | 9 (2.9) | |
| **Other** | | | |
| No | 421 (61.9) | 207 (66.6) | 0.16 |
| Yes | 259 (38.1) | 104 (33.4) | |
| **Operating protocol in the woman's file** | | | < 0.0001 |
| Complete, and mentions the procedure in detail along with all post-operative instructions | 452 (66.8) | 293 (86.2) | |
| Incomplete, only mentions the procedure but in detail | 22 (3.3) | 31 (9.1) | |
| Incomplete, only gives a summary of the procedure | 196 (28.9) | 16 (4.7) | |
| None | 7 (1.0) | 0.0 | |
| **Type of incision** | | | < 0.0001 |
| No | 18 (2.6) | 5 (1.4) | |
| Transverse | 296 (42.7) | 200 (56.2) | |
| Midline | 380 (54.7) | 151 (42.4) | |
| **Transfusion performed** | | | 0.04 |
| Yes | 37 (5.7) | 10 (2.8) | |
| No | 609 (94.3) | 343 (97.2) | |
| **Qualification of the person who performed the caesarean** | | | 0.015 |
| General practitioner | 631 (93.8) | 320 (91.7) | |
| Surgeon | 10 (1.5) | 1 (0.3) | |
| Gynaecologist | 32 (4.8) | 26 (7.5) | |
| General nurse | 0 | 2 (0.6) | |

## Pregnancy outcomes

Our study shows more cases of low birth weight, more post-operative complications, more maternal deaths, lower perinatal mortality, shorter stays in hospital and more complete patient files in group 2 than in group 1 (Table 4).

## Logistic regression

The association of risk factors for low birth weight is due to health staff training OR 1.7 (IC95% 1.1–2.6), primary residence (rural location) OR 2.4 (IC95% 1.5–3.8), type of admission OR 0.33 (IC95% 0.14–0.80), fundal height before CS OR 0.74 (IC95% 0.68–0.80), presentation of the foetus OR 2.1 (breech), OR 5.1 (other presentation), and type of anaesthesia, particularly general anaesthesia OR 1.7 (IC95% 1.1–2.6) (Table 5).

Table 4. Pregnancy outcomes.

| | No Intervention (n = 694) | Intervention (n = 356) | p-value |
|---|---|---|---|
| **APGAR at 1 min** | | | *0.17* |
| ≥ 7 | 589 (86.9) | 318 (89.8) | |
| < 7 | 89 (13.1) | 36 (10.2) | |
| **APGAR at 5 min** | | | *0.36* |
| ≥7 | 639 (94.4) | 330 (93) | |
| < 7 | 38 (5.6) | 25 (7) | |
| **APGAR at 10 min** | | | *0.49* |
| ≥7 | 646 (95.4) | 342 (96.3) | |
| < 7 | 31 (4.6) | 13 (3.7) | |
| **Baby's birth weight in grams** | | | *0.02* |
| ≥ 2500g | 553 (92.3) | 309 (87.8) | |
| < 2500 g | 46 (7.7) | 43 (12.2) | |
| **Intra-operative complications** | | | |
| No | 653 (97.6) | 338 (97.4) | *0.84* |
| Yes | 16 (2.4) | 9 (2.6) | |
| **Post-operative complications** | | | |
| No | 651 (97.6) | 16 (4.7) | *< 0.0001* |
| Yes | 16 (2.4) | 326 (95.3) | |
| **Mother's outcome** | | | *0.046* |
| Death | 1 (0.15) | 4 (1.2) | |
| Discharged | 674 (99.7) | 337 (98.8) | |
| Transferred | 1 (0.15) | 0 | |
| **Infant's outcome** | | | *0.04* |
| Discharged | 636 (94.1) | 334 (96.8) | |
| Macerated stillbirth | 11 (1.6) | 1 (0.3) | |
| Fresh stillbirth | 17 (2.5) | 3 (0.9) | |
| Unspecified stillbirth | 2 (0.3) | 3 (0.9) | |
| Live birth but died within 24h | 3 (0.4) | 3 (0.9) | |
| Transferred | 7 (1.0) | 1 (0.3) | |
| **Neonatal death** | | | |
| No | 643 (95.1) | 335 (97.1) | *0.14* |
| Yes | 33 (4.9) | 10 (2.9) | |
| **Total duration of stay in days** | | | *< 0.0001* |
| | 5 (5–7) | *3 (3–7)* | |
| **Patient's file is complete** | | | *< 0.0001* |
| Yes | 360 (53) | 249 (92.9) | |
| No | 319 (47) | 19 (7.1) | |

The association of risk factors for neonatal mortality were non-cephalic presentation of the foetus OR 1.2, 4.4, the use of general anaesthesia OR 3.6 (IC95% 1.9–6.8), and indications for CS other than dystocia (foetal distress, placenta praevia, scarred uterus) OR 0.14 (IC95% 0.02–1.03) (Table 6).

Our study showed more post-operative complications among single women OR 1.8 (IC95% 1.3–2.7), multigravidas OR 2.1 and 1.9, women who had a history of at least two prior CSs OR 3.3 (IC95% 2.3–4.6), or those who had a job (OR 4.8) or a low level of education (OR 3.5, 4), living in a rural area or far from the hospital in the health zone of their village of origin OR 7 (IC95% 4.9–10). Non-cephalic presentation of the fetus (OR 0.86, 2.6), a CS prior to

**Table 5. Univariate and multiple logistic regression of the association of risk factors for low birth weight.**

| | OR (IC 95%) | *p-value* | aOR (IC 95%) | *p-value* |
|---|---|---|---|---|
| **Intervention** | | | | |
| No | 1 | *0.02* | 1 | *0.13* |
| Yes | 1.7 (1.1–2.6) | | 1.9 (0.8–4.3) | |
| **Woman's age in years** | 1.01 (0.97–1.05) | *0.57* | | |
| **Marital status** | | | | |
| Married | 1 | *0.92* | | |
| Single | 1.03 (0.54–1.96) | | | |
| **Woman's level of education** | | | | |
| None | 3.2 (0.91–11.4) | *0.09* | | |
| Primary | 3.4 (0.95–12.0) | | | |
| Secondary | 2.9 (0.86–9.6) | | | |
| Higher | 1 | | | |
| Other | 5.3 (1.5–18.4) | | | |
| **Woman's occupation** | | | | |
| Housewife | 1 | *0.49* | | |
| Job | 1.2 (0.71–2.1) | | | |
| Other | 0.73 (0.34–1.57) | | | |
| **Primary residence** | | | | |
| Urban area | 1 | *< 0.0001* | 1 | *0.55* |
| Rural area | 2.4 (1.5–3.8) | | 1.2 (0.61–2.5) | |
| **Parity** | | | | |
| 0 | 1 | *0.93* | | |
| 1–3 | 1.1 (0.63–1.87) | | | |
| ≥ 4 | 1.1 (0.60–2.1) | | | |
| **Number of previous caesareans** | | | | |
| 0 | 1 | *0.49* | | |
| 1 | 0.80 (0.43–1.48) | | | |
| ≥2 | 1.2 (0.69–2.05) | | | |
| **Hospital in the health zone of the woman's village of origin** | | | | |
| Yes | 1 | *0.37* | | |
| No | 1.2 (0.78–1.96) | | | |
| **Type of admission** | | | | |
| Normal prior to onset of labour | 1 | | | |
| Emergency during labour | 0.24 (0.8–2.09) | | | |
| **Mode of admission** | | | | |
| Referral prior to onset of labour | 1 | *0.04* | 1 | *0.90* |
| Referral during labour | 0.69 (0.38–1.24) | | 0.85 (0.29–2.47) | |
| Personal decision prior to onset of labour | 0.33 (0.14–0.80) | | 0.96 (0.44–2.1) | |
| Personal decision during labour | 0.51 (0.28–0.93) | | 0.73 (0.30–1.79) | |
| **Fundal height in centimetres before caesarean** | | *< 0.0001* | | |
| | 0.74 (0.68–0.80) | | 0.73 (0.67–0.80) | *< 0.0001* |
| **Presentation of the foetus** | | | | |
| Cephalic | 1 | *0.0001* | 1 | *0.004* |
| Breech | 2.1 (1.04–4.4) | | 2.6 (1.15–5.91) | |
| Other | 5.1 (2.2–11.6) | | 4.1 (1.4–12.2) | |
| **Onset of labour** | | | | |

(*Continued*)

**Table 5.** (Continued)

| | OR (IC 95%) | p-value | aOR (IC 95%) | p-value |
|---|---|---|---|---|
| Spontaneous | 1 | 0.35 | | |
| Induced | 0.9 (0.43–1.74) | | | |
| Caesarean before labour | 1.6 (0.80–3.15) | | | |
| **Type of primary anaesthesia** | | | | |
| **loco-regional** | 1 | 0.03 | 1 | 0.01 |
| **general** | 1.7 (1.1–2.6) | | 1.6 (0.86–2.8) | |
| **Pelvic anomaly** | | | | |
| No | 1 | 0.08 | | |
| yes | 0.50 (0.22–1.10) | | | |
| **Post-operative indication for Caesarean 1** | | | | |
| **Pelvic anomaly** | | | | |
| No | 1 | 0.08 | | |
| yes | 0.49 (0.22–1.09) | | | |
| **Scarred uterus** | | | | |
| No | 1 | 0.06 | | |
| yes | 0.58 (0.32–1.03) | | | |

onset of labour OR 3 (IC95% 1.9–4.8) and indications such as dystocia are associated with more post-operative complications OR 0.48 (IC95% 0.28–0.67) (Table 7).

## Discussion

### Caesarean frequency

Our study shows an average CS proportion of 17%, which is in line with the average rate of 10 to 15% recommended by the WHO [16]. These WHO recommendations are valid particularly for scheduled caesarean sections, for which the proportion should be very low except in centers for high-risk pregnancies. The CS proportion seen in our study is higher than that seen among the general population in the Democratic Republic of Congo (DRC) [9]. One study analysed the proportion of CS in SSA and showed an average CS rate of 19% [17], which is in line with the overall estimate described in the literature [18]. Another study on the analysis of CS practices in SSA described a CS ranging from 2% to 52% [19]. This study showed that the rate varies depending on the population studied and on access to healthcare [19]. The proportion of CS is higher in both groups at the North Kivu provincial hospital, which is a skills development center for health staff in the province and as such receives more referrals from outlying facilities. The PADISS project (to support the health system's integrated development), developed by ULB (Universitélibre de Bruxelles) Cooperation and Erasme Cooperation alongside the local authorities, is firmly established in the North Kivu health system and is working simultaneously on different key elements that make it possible to offer the population higher quality, consistent, effective healthcare. Its objective is to improve the quality and accessibility of healthcare in the province of North Kivu and to ensure its stability by gradually putting in place, in line with the contractual subsidy scheme which is being implemented in the province, a system for the accreditation of health facilities and staff as well as the necessary elements for their proper operation. The presence of this project explains the higher CS frequency at North Kivu provincial hospital. Although a CS is an effective technique for preventing maternal and perinatal mortality when used appropriately, it is not risk free and is associated with short- and long-term complications [20]. These rates have risen in developed and

**Table 6. Univariate and multiple logistic regression of the association of risk factors for neonatal mortality.**

| | OR (IC 95%) | *p-value* | aOR (IC 95%) | *p-value* |
|---|---|---|---|---|
| **Intervention** | | | | |
| No | 1 | *0.14* | 1 | *0.21* |
| Yes | 0.58 (0.28–1.19) | | 0.69 (0.28–1.33) | |
| **Woman'sage in years** | 1.03 (0.98–1.09) | *0.20* | | |
| **Marital status** | | | | |
| Married | 1 | *0.69* | | |
| Single | 0.8 (0.32–2.13) | | | |
| **Woman'slevel of education** | | | | |
| None | 1.7 (0.44–6.6) | *0.24* | | |
| Primary | 0.24 (0.03–2.4) | | | |
| Secondary | 1.7 (0.48–5.8) | | | |
| Higher | 1 | | | |
| Other | 2.4 (0.63–9.4) | | | |
| **Woman's occupation** | | | | |
| Housewife | 1 | *0.52* | | |
| Job | 0.76 (0.31–1.9) | | | |
| Other | 1.4 (0.62–3.4) | | | |
| **Primary residence** | | | | |
| Urban area | 1 | *0.46* | | |
| Rural area | 1.3 (0.62–2.8) | | | |
| **Parity** | | | | |
| 0 | 1 | *0.11* | | |
| 1–3 | 2.8 (1.0–7.3) | | | |
| ≥ 4 | 2.7 (0.93–7.7) | | | |
| **Number of previous caesareans** | | | | |
| 0 | 1 | *0.57* | | |
| 1 | 0.73 (0.30–1.80) | | | |
| ≥ 2 | 1.22 (0.57–2.62) | | | |
| **Hospital in the health zone of the woman's village of origin** | | | | |
| Yes | 1 | *0.33* | | |
| No | 1.4 (0.72–2.64) | | | |
| **Type of admission** | | | | |
| Normal prior to onset of labour | | | | |
| Emergency during labour | | | | |
| **Mode of admission** | | | | |
| Referral prior to onset of labour | 1 | *0.47* | | |
| Referral during labour | 1.1 (0.42–2.98) | | | |
| Personal decision prior to onset of labour | 0.34 (0.07–1.69) | | | |
| Personal decision during labour | 1.03 (0.40–2.65) | | | |
| **Fundal height in centimetres before caesarean** | | | | |
| | 0.96 (0.87–1.05) | *0.38* | | |
| **Presentation of the foetus** | | | | |
| Cephalic | 1 | *0.03* | 1 | *0.007* |
| Breech | 1.2 (0.36–4.0) | | 0.75 (0.17–3.2) | |
| Other | 4.4 (1.4–13.3) | | 6.3 (2.0–20) | |
| **Onset of labour** | | | | |

(*Continued*)

**Table 6.** (Continued)

| | OR (IC 95%) | *p-value* | aOR (IC 95%) | *p-value* |
|---|---|---|---|---|
| Spontaneous | 1 | *0.06* | | |
| Induced | 2.5 (1.2–5.3) | | | |
| Caesarean before labour | 1.5 (0.50–4.3) | | | |
| **Type of primary anaesthesia** | | | | |
| Loco-regional | 1 | *< 0.0001* | 1 | *0.001* |
| General | 3.6 (1.9–6.8) | | 3.1 (1.6–6.2) | |
| **Pre-operative indication for Caesarean 1** | | | | |
| **Pelvic anomaly** | | | | |
| No | 1 | *0.05* | | |
| Yes | 0.14 (0.02–1.03) | | | |
| **Post-operative indication for Caesarean 1** | | | | |
| **Pelvic anomaly** | | | | |
| No | 1 | *0.05* | | |
| Yes | 0.14 (0.02–1.00) | | | |
| **Scarred uterus** | | | | |
| No | 1 | *0.54* | | |
| Yes | 0.79 (0.37–1.68) | | | |

developing countries alike, sometimes reaching very high rates as, for example, in Brazil and the United States [21].

## Indications for caesarean

For both groups, approximately 62% of the caesarean sections had been performed as an emergency during labour. The emergency CS rate in both groups is as high as the rates given in the literature for the DRC and other regions in developing countries, which report emergency CS rates ranging from 58% to 98%. Some reasons for these high CS rates include the absence of quality antenatal care which can prophylactically detect and refer high-risk pregnancies toward specialist facilities, the dramatic rise in makeshift and uncertified maternity units, the low level of qualification of health staff working in these maternity units, lack of awareness of the counter-indications for vaginal birth followed by delayed transfer to specialist facilities, poor distribution of health centers and difficult access to referral facilities, as well as poverty and illiteracy among these populations [10, 22]. These are common factors and characteristics, in varying degrees, of developing countries. Although the CS had been performed more as an emergency in both studies, the majority of patients had been referred before and during labour to the better equipped hospital, labour was more often spontaneous, there were fewer cases of induced labour and the CS was performed more frequently prior to labour in the study after the staff training. The main indications for CS in both studies were dystocia, foetal distress and a scarred uterus.

Dystocia (difficult and protracted labour) was the primary indication for CS in both groups. Dystocia is mainly caused by insufficient uterine contractions, sometimes due to cephalopelvic disproportion, lack of progress in foetal descent due to a tumour; however, it is sometimes difficult to make this diagnosis prior to labour [19]. Certain authors wondered whether dystocia was being over-diagnosed nowadays, in order to justify more frequent use of CS [11]. Our study shows fewer indications of dystocia after training than before (p = 0.005).

**Table 7.** *Univariate* logistic regression of the association of risk factors for post-operative complications.

| | Complications n(%) | OR (IC 95%) | p-value |
|---|---|---|---|
| **Woman'sage in years** | | 0.99 (0.97–1.02) | 0.56 |
| **Marital status** | | | |
| Married | 271 (31.8) | 1 | 0.001 |
| Single | 60 (46.2) | 1.8 (1.3–2.7) | |
| **Woman'slevel of education** | | | |
| None | 95 (53.4) | 3.5 (2.1–6.0) | < 0.0001 |
| Primary | 85 (56.3) | 4.0 (2.3–6.8) | |
| Secondary | 122 (30.1) | 1.3 (0.81–2.**16**) | |
| Higher | 26 (24.5) | 1 | |
| Other | 5 (3.9) | 0.12 (0.05–0.33) | |
| **Woman's occupation** | | | |
| Housewife | 185 (26.7) | 1 | < 0.0001 |
| Job | 116 (63.4) | 4.8 (3.4–6.7) | |
| Other | 38 (32.5) | 1.3 (0.87–2.0) | |
| **Primary residence** | | | |
| Urban area | 213 (26.5) | 1 | < 0.0001 |
| Rural area | 129 (71.7) | 7.0 (4.9–10) | |
| **Parity** | | | |
| 0 | 61 (23.1) | 1 | 0.0001 |
| 1–3 | 193 (39.1) | 2.1 (1.5–3.0) | |
| $\geq 4$ | 88 (35.9) | 1.9 (1.3–2.7) | |
| **Number of previous caesareans** | | | |
| 0 | 96 (23.9) | 1 | < 0.0001 |
| 1 | 82 (35.7) | 1.8 (1.2–2.5) | |
| $\geq 2$ | 12 ((50.6) | 3.3 (2.3–4.6) | |
| **Hospital in the health zone of the woman's village of origin** | | | |
| Yes | 160 (29.1) | 1 | 0.001 |
| No | 156 (39.1) | 1.6 (1.2–2.1) | |
| **Type of admission** | | | |
| Normal prior to onset of labour | | | |
| Emergency during labour | | | |
| **Fundal height in centimetres before caesarean** | | | |
| | | 1.04 (1.00–1.08) | 0.05 |
| **Presentation of the foetus** | | | |
| Cephalic | 300 (33.6) | 1 | 0.03 |
| Breech | 21 (30.4) | 0.86 (0.51–1.47) | |
| Other | 17 (56.7) | 2.6 (1.2–5.4) | |
| **Onset of labour** | | | |
| Spontaneous | 274 (35.5) | 1 | < 0.0001 |
| Induced | 11 (8.5) | 0.17 (0.09–0.32) | |
| Caesarean before labour | 51 (62.2) | 3.0 (1.9–4.8) | |
| **Type of primary anaesthesia** | | | |
| Loco-regional | 241 (36.1) | 1 | 0.04 |
| General | 98 (29.6) | 0.74 (0.56–0.99) | |
| **Pre-operative indication for Caesarean 1** | | | |
| **Pelvic anomaly** | | | |
| No | 305 (35.8) | 1 | < 0.0001 |
| Yes | 28 (19.4) | 0.43 (0.28–0.67) | |

A scarred uterus was one of the primary indications for caesarean sections. The "once a cae-sarean always a caesarean" policy is widely applied in SSA, mainly from fear of uterine rupture during labour. This policy helps reduce both the uterine rupture rate and the emergency sur-gery responsible for the increase in maternal and perinatal mortality and morbidity [19]. These repeat caesareans do not, however, result in the medical benefits expected. In fact, a vag-inal delivery after a caesarean section has a low risk both for the mother and for the child [23]. The risk of uterine rupture increases with the number of previous CS and our study shows that a CS was most often performed on a multi-scarred uterus after the training than before (p < 0.0001).

Foetal distress was one of the main indications for CS in our study. The accuracy of this diagnosis is sometimes doubtful. Foetal monitoring is not yet used continuously in many of these medical facilities, and health staff still need to be trained in its interpretation. One study on foetal monitoring was unable to show an improvement in the infants' well-being parame-ters in relation to the use of a Pinard stethoscope [24]. However, this study showed an increase in the use of caesarean section when monitoring was used, because it is difficult to make the distinction between foetal stress and true distress, as shown by these authors [25]. Monitoring gives a high false positive rate for foetal distress: as such, it is advisable to use ST analysis (STAN) or scalp pH, which is non-existent in these medical facilities. It would be worthwhile setting up foetal monitoring and ST analysis or scalp pH in these different facilities in order to reduce this false positive rate along with the CS rate for foetal distress. In the DRC, one study determined foetal distress as a pre-operative indication for CS in 23% of cases in urban settings in university clinics in Kinshasa, whereas in the same period, the rates reported in semi-urban settings (Mbuji-Mayi) and rural settings in the same country were 1.5% and 0% respectively [26]. This rate of 23% in urban settings is higher than that described in our study. One expla-nation for this is that foetal distress is correctly diagnosed in university hospital settings where more labour monitoring equipment is available. Our study showed fewer diagnoses of foetal distress after training than before, but this difference was not statistically significant (p = 0.11).

Other indications such as haemorrhagic placenta praevia, uterine pre-rupture and rupture, and breech, brow and transverse presentation explain the high rate of emergency CS in our study.

## Maternal risks

Intraoperative complications were approximately 2.5% in both groups; however, post-opera-tive complications were higher in group 2 (p < 0.0001), explained by the fact that the patient's file was more complete (p < 0.0001), and more incisions were transverse (p < 0.0001) in group 2. The staff who performed the CS were more qualified in group 2 (p = 0.015). Accord-ing to the Cochrane systematic review, the Joel-Cohen incision has advantages over the sub umbilical midline incision, resulting in fewer cases of fever, pain, need for analgesics, blood loss, and a shorter procedure and hospitalization duration. However, these studies do not pro-vide information relating to mortality and severe or long-term morbidity [27]. Maternal mor-tality was higher in group 2 (p = 0.046) due to the fact that the patients' files were more complete, deaths were properly recorded, and the training made it possible to select at-risk cases giving a high maternal mortality, but which remains low in relation to several regions in SSA. This decrease in mortality may be explained by the fact that patients were not monitored up to 42 days post-partum, because the average stay in hospital was five days. While CS is safer in developed countries, it still entails the risks of many major abdominal procedures in the DRC. Maternal mortality is estimated to be approximately 2 to 11 times higher after a CS than after a vaginal delivery [28]. There were fewer blood transfusions performed after the training

(p = 0.04), explained by the fact that several risk factors for anaemia had been identified during the antenatal visits, of which there were more after the training.

### Perinatal risks

Our study describes fewer neonatal deaths after the training than before, but this difference is not statistically significant (p = 0.14). This neonatal mortality was primarily associated with the use of general anaesthesia. One retrospective study gave a newborn infant death rate of 9% after CS or an APGAR score lower than 7 at five minutes after birth [29]. This rate was comparable with the results described in Africa where a WHO study showed an average neonatal mortality rate of 12.9% after a CS [6]. Our findings on mortality rates were lower than those described by other studies [6, 30]. The use of general anaesthesia shows that the majority of CS had been performed as an emergency and were accompanied by more maternal and perinatal risks as shown by these authors [31]. Neonatal mortality was higher among multiparous patients as described in other studies conducted in Rwanda and Nigeria [32]. One possible explanation for this is that women who have more children are often poor and less educated [32]. Poverty and a low level of education have been associated with poor neonatal outcomes in SSA [30]. Our study shows more cases of low birth weight after training than before (p < 0.002). The association of risk factors for low birth weight is related to health staff training, primary residence, mode of admission, fundal height before the CS, non-cephalic presentation of the foetus, and the use of general anaesthesia. However, we recommend continuing efforts to raise awareness among pregnant women as to the value of antenatal visits with a view to decreasing the maternal and perinatal risks shown in our results and those of other studies [33].

There are several limitations to take into account in this comparative study. Given the cross-cutting nature of the two studies, and certain shortcomings in medical practice, several variables were lacking data in the retrospective study, such as size, weight for calculating the body mass index, gestational age at the time of the CS, and a partograph in the medical file in order to assess the diagnosis of dystocia or dyskinesia. This study was conducted in hospital facilities in Goma, and does not therefore reflect the situation in rural areas of North Kivu and the rest of the DRC.

### Conclusion

Caesarean section should be a factor in reducing foeto-maternal morbidity and mortality if the transfer conditions, the working conditions at referral center level, and staff training are improved. Our study showed more post-operative complications, more maternal deaths, more cases of scarred uterus, more low-birth weight infants, fewer indications of foetal distress and dystocia, and fewer neonatal deaths in group 2. The PADISS project will help to improve the maternal and perinatal risks seen in this comparative study. It may be necessary to improve the conditions for transferring patients to referral hospitals, the training of health staff involved in antenatal visits in order to ensure timely detection and referral of high-risk pregnancies to referral hospitals which must have adequate equipment for providing quality emergency caesarean sections and neonatal care in order to reduce the perinatal risks associated with CS. A wide-scale prospective study may be necessary to assess the impact of the training on improving the quality of CS procedures in North Kivu and the Democratic Republic of Congo.

### Supporting information

**S1 Data.**
(XLSX)

**S2 Data.**

(XLS)

## Author Contributions

**Conceptualization:** Michel Dikete Ekanga, Prudence Mitangala, Judith Racape.

**Data curation:** Michel Dikete Ekanga, Prudence Mitangala, Yves Coppieters, Richard Kabuyanga Kabuseba, Judith Racape.

**Formal analysis:** Michel Dikete Ekanga, Prudence Mitangala, Yves Coppieters, Christine Kirkpatrick, Philippe Simon, Judith Racape, Wei-Hong Zang.

**Funding acquisition:** Michel Dikete Ekanga, Judith Racape.

**Investigation:** Michel Dikete Ekanga, Christine Kirkpatrick, Judith Racape, Wei-Hong Zang.

**Methodology:** Michel Dikete Ekanga, Yves Coppieters, Wei-Hong Zang.

**Project administration:** Michel Dikete Ekanga, Yvon Englert.

**Resources:** Michel Dikete Ekanga.

**Software:** Michel Dikete Ekanga.

**Supervision:** Michel Dikete Ekanga, Yves Coppieters, Philippe Simon.

**Validation:** Michel Dikete Ekanga, Yves Coppieters, Christine Kirkpatrick, Richard Kabuyanga Kabuseba, Philippe Simon, Judith Racape, Wei-Hong Zang.

**Visualization:** Michel Dikete Ekanga, Christine Kirkpatrick.

**Writing – original draft:** Michel Dikete Ekanga.

**Writing – review & editing:** Michel Dikete Ekanga.

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
