## [Decision Letter · Decision Letter 0]

10 Aug 2021

PONE-D-20-41094

EVOLUTION IN CAESAREAN SECTION PRATICES IN NORTH-KIVU; IMPACT OF CAREGIVER TRAINING

PLOS ONE

Dear Dr. dikete ekanga,

Thank you for submitting your manuscript to PLOS ONE. After careful consideration, we feel that it has merit but does not fully meet PLOS ONE’s publication criteria as it currently stands. Therefore, we invite you to submit a revised version of the manuscript that addresses the points raised during the review process.

The reviewer has requested some additions and revisions, in addition to the items raised by the reviewer, please address the following points before more consideration:

It appears the authors must use multiple regression (only one dependent variable in each model) instead of multivariate (more than two dependent variables).

Please report regression prerequisite in method section and its handling methods.

Provide B: Unstandardized coefficients, S.E.: Standard error and Beta: Standardized regression coefficients for variable and model Goodness of Fit in regression table.

Please present abbreviation (OR, AOR, ….) in their full format first time.

It appears in regression analysis reported in tables’ 5-7 only women with selected condition (only women’s with low birth weight, only women’s with neonatal mortality and only women’s with postoperative complications) considered as cases and were included in modeling and other cases were excluded. In regression analysis your dependent variable is women’s condition in relation to selected variables and group (Intervention – and Intervention -+) is your independent variable. And other factors were confounding factors. Statistical analysis must be reviewed and corrected before more consideration. 

In intervention section you declared “Following these findings, specific training was given to the local health staff.”, but we do not have any findings before this section.

Provide your ethic statement in method section and ethic committee approval form as appendix.

In addition, the reviewers request improvements to the clarity of the written language.

We look forward to receiving your revised manuscript.

Kind regards,

Kamal Gholipour, PhD

Academic Editor

PLOS ONE

Journal Requirements:

2. Please provide additional details regarding participant consent for your prospective study. In the ethics statement in the Methods and online submission information, please ensure that you have specified (1) whether consent was informed and (2) what type you obtained (for instance, written or verbal, and if verbal, how it was documented and witnessed). If your study included minors, state whether you obtained consent from parents or guardians. If the need for consent was waived by the ethics committee, please include this information.

3. In your ethics statement in the manuscript and in the online submission form, please ensure that you have discussed whether all data/samples were fully anonymized before you accessed them and/or whether the IRB or ethics committee waived the requirement for informed consent for the retrospective study. If patients provided informed written consent to have data/samples from their medical records used in research, please include this information.

4. In the ethics statement in the manuscript and in the online submission form, please provide additional information about the patient records/samples used in your retrospective study, including the date range (month and year) during which patients' medical records/samples were accessed.

"This research work has not been funded by any organization. It is a research work in collaboration with the Free University of Brussels and the Erasmus Brussels Academic Hospital"

7. PLOS requires an ORCID iD for the corresponding author in Editorial Manager on papers submitted after December 6th, 2016. Please ensure that you have an ORCID iD and that it is validated in Editorial Manager. To do this, go to ‘Update my Information’ (in the upper left-hand corner of the main menu), and click on the Fetch/Validate link next to the ORCID field. This will take you to the ORCID site and allow you to create a new iD or authenticate a pre-existing iD in Editorial Manager. Please see the following video for instructions on linking an ORCID iD to your Editorial Manager account: https://www.youtube.com/watch?v=_xcclfuvtxQ.

8. Please include your full ethics statement in the ‘Methods’ section of your manuscript file. In your statement, please include the full name of the IRB or ethics committee who approved or waived your study, as well as whether or not you obtained informed written or verbal consent. If consent was waived for your study, please include this information in your statement as well. 

9. We note you have included a table to which you do not refer in the text of your manuscript. Please ensure that you refer to Table 4 in your text; if accepted, production will need this reference to link the reader to the Table.

10. Please include your tables as part of your main manuscript and remove the individual files. Please note that supplementary tables (should remain/ be uploaded) as separate "supporting information" files.

Reviewers' comments:

Reviewer's Responses to Questions

**Comments to the Author**

1. Is the manuscript technically sound, and do the data support the conclusions?

Reviewer #1: Partly

Reviewer #2: Yes

Reviewer #3: Partly

2. Has the statistical analysis been performed appropriately and rigorously? 

Reviewer #1: Yes

Reviewer #2: Yes

Reviewer #3: Yes

3. Have the authors made all data underlying the findings in their manuscript fully available?

Reviewer #1: Yes

Reviewer #2: Yes

Reviewer #3: Yes

4. Is the manuscript presented in an intelligible fashion and written in standard English?

Reviewer #1: Yes

Reviewer #2: Yes

Reviewer #3: Yes

5. Review Comments to the Author

Reviewer #1: The objective of this study was to determine the impact of health staff training in a region of the DRC and which focused on the quality of C-section procedures by comparing outcomes determined at 2 sequential time points.

Multivariate logistic regression is used to determine which characteristics were associated with maternal and perinatal mortality and morbidity.

It is unlikely that dystocia or true fetal distress rates would be directly related to training, rather changes may be a reflection on how these diagnoses were defined and reported. A reduction in neonatal deaths is a “hard” indicator and maybe associated with the quality of training provided. The increase in scarred uteri may simply reflect a relative increase in the total number of prior cesarean sections. While training is relevant to ensure safe surgery practices for this group of women, in order to maximize study findings, it may be best to focus on outcomes among nulliparous women. Low birth weight is associated with a number of direct and indirect causes including access to adequate nutrition and the presence of micronutrient deficiencies, especially that of iron and folic acid.

Validated training and implementation of training procedures will likely not be a direct cause of untoward clinical outcomes. However, fetal distress is often cited as the reason for surgical intervention.

Cesarean sections performed in sub-Saharan African countries, while often indicated, are associated with higher rates of poor outcomes than those reported from other regions. The 3 Delays, including time to reach a facility as well as training and facility infrastructure must be addressed in future studies.

A few questions:

Was there a reduction in fistulas or reports of incontinence?

How often was the WHO partogram used?

Was there a difference in outcome between non-emergent c-section patients as compared to the two thirds of women who were considered emergencies?

During the second phase of this study, was more electronic monitoring equipment available?

Was there a report by the ERASMUS training team on their observations relative to the overall quality of care and challenges that need to be addressed?

There has been a paucity of papers related to cesarean section rates and outcomes in the DRC. While it is difficult to draw meaningful conclusions from the data presented, the paper is still an important contribution in demonstrating the difficulties of addressing composite training needs in the face of limited resources and a challenging topography.

Reviewer #2: Unvariate and multivariate logistic regression was performed to determine which characteristics are associated with maternal and perinatal morbidity and mortality.These research methods are scientific and reasonable.

However, there are still some problems in this paper:

Baseline population was recruited from referral hospitals in the city of Goma from the period 01/11/2013 to 01/01/2016 (group 1).The second group was recruited from the same hospitals in Goma for the period from June 2019 to January 2020 (group 2). The time period of the two groups of people is not consistent, which may have an impact on the comparison results. It is suggested to explain this in the discussion or in the deficiencies of this paper.

The writing of the article should be standardized. For example, all tables should be three-line tables.The article lacks key words.

Reviewer #3: 1. In the section of result，authors concluded that “Our study shows more cases of low birth weight, more post-operative complications, more maternal deaths, lower perinatal mortality… in group 2(after intervention) than in group 1(before intervention).”(as shown in Table 4). However, multivariate logistic regression did not support the conclusion. As shown in Table 5, intervention had no statistically significant impact on low birth weight (OR=1.9, P=0.13). As shown in Table 6, intervention had no statistically significant impact on neonatal mortality (OR=0.69, P=0.21). Given the fact that there are significant differences in socio-demographic characteristics, medical data and obstetric characteristics between the two groups, multivariate logistic regression result is more reliable. Also, the result of multivariate logistic regression should be added in the abstract.

2. Is it reasonable to choose low birth weight, maternal deaths, and perinatal mortality as the evaluable indicator? The objective of the study is to assess the impact of health staff training on the quality of CS procedures. Firstly, low birth weight cannot reflect the quality of CS procedures very well; Secondly, the incidence rate of maternal deaths, and perinatal mortality is very low, it is hard to measure a significant difference (unless the sample size is very large). Maybe the indicator such as operative time, blood loss, APGAR, duration of stay and post-operative complications are more suitable.

3. APGAR and duration of stay were compared between the two groups in the study, but the result did not described in the abstract. Although there are no statistical significance in APGAR between the two groups by Chi² Pearson test, a multivariate logistic regression should be added to assess the impact of intervention on APGAR.

4. In table 7, the variable “intervention” should be added. Also, multivariate logistic regression result should be added to assess the impact of intervention on post-operative complications.

5. In the study, statistical analysis is performed appropriately, but the interpretation of the statistical results requires improvement. In the abstract, “Our study shows a decrease in neonatal deaths, dystocia and foetal distress, but an increase in post-operative complications, maternal deaths and cases of scarred uterus and low birth weight.” Among them, dystocia, foetal distress and scarred uterus is the obstetric characteristics of maternal, which can hardly be influenced by health staff training and cannot reflect the quality of CS procedures very well.

6. Some data should be checked. For example, the data of post-operative complications in table 4 should be checked.

7. In the abstract “In the population studied prospectively, after the implementation of health staff training, there were fewer cases of dystocia, foetal distress and neonatal death, but more cases of scarred uterus, post-operative complications and low birth weight.” The word “cases” is not used properly. Maybe “incidence rate” is more suitable.

6. PLOS authors have the option to publish the peer review history of their article (what does this mean?). If published, this will include your full peer review and any attached files.

Reviewer #1: No

Reviewer #2: No

Reviewer #3: No

---

## [Author Response · Author response to Decision Letter 0]

21 Sep 2021

After thorough analysis of all the comments made to this document, we sincerely thank the 3 reviewers for their time and also the way in which the article was corrected. We can only encourage other colleagues and ourselves to submit research to Plos One.

---

## [Decision Letter · Decision Letter 1]

8 Nov 2021

PONE-D-20-41094R1EVOLUTION IN CAESAREAN SECTION PRATICES IN NORTH-KIVU; IMPACT OF CAREGIVER TRAININGPLOS ONE

Dear Dr. dikete ekanga,

Thank you for submitting your manuscript to PLOS ONE. After careful consideration, we feel that it has merit but does not fully meet PLOS ONE’s publication criteria as it currently stands. Therefore, we invite you to submit a revised version of the manuscript that addresses the points raised during the review process.

The authors did a nice job addressing comments and suggestions. The reviewers find the work of merit but we have requested some additions and revisions, please address the following points before final decision.==============================

We look forward to receiving your revised manuscript.

Kind regards,

Kamal Gholipour, PhD

Academic Editor

PLOS ONE

Journal Requirements:

Reviewers' comments:

Reviewer's Responses to Questions

**Comments to the Author**

1. If the authors have adequately addressed your comments raised in a previous round of review and you feel that this manuscript is now acceptable for publication, you may indicate that here to bypass the “Comments to the Author” section, enter your conflict of interest statement in the “Confidential to Editor” section, and submit your "Accept" recommendation.

Reviewer #1: All comments have been addressed

Reviewer #2: (No Response)

Reviewer #3: (No Response)

2. Is the manuscript technically sound, and do the data support the conclusions?

Reviewer #1: Yes

Reviewer #2: Yes

Reviewer #3: Partly

3. Has the statistical analysis been performed appropriately and rigorously? 

Reviewer #1: Yes

Reviewer #2: Yes

Reviewer #3: Yes

4. Have the authors made all data underlying the findings in their manuscript fully available?

Reviewer #1: Yes

Reviewer #2: Yes

Reviewer #3: Yes

5. Is the manuscript presented in an intelligible fashion and written in standard English?

Reviewer #1: Yes

Reviewer #2: Yes

Reviewer #3: Yes

6. Review Comments to the Author

Reviewer #1: Globally, cesarean section continues to remain on the rise, and in some cases have led to improvement in maternal and neonatal outcomes. Unfortunately, such is not the case in much of sub-Saharan Africa, not obvious from the data reported in this paper where there was no neonatal benefit and 1 in 500 women died perhaps as a result of this procedure. Undoubtedly, the death rates were even greater, as few women present for a postpartum visits, yet it is in this period that many cases of postpartum sepsis and its sequelae are to be found.

A comparison of complications among two groups of women based upon the year they delivered was well described. These were a heterogeneous grouping with significant differences in baseline characteristics.

Poor outcomes linked to cesarean section births are multifactorial and the authors correctly point out areas that are not easily addressable without a major infusion of resources.

A few points are worth noting:

1. The data presented is primarily from a tertiary care (transfer) facility. Given the difficult geography of the DRC, it is not surprising that by the time a woman reaches a tertiary care hospital, many of the cesarean sections are characterized as urgent – a designation associated with much poorer outcomes.

2. Rates of cesarean section in many parts of sub-Saharan Africa have remained at 4-5%. With 17-19% cesarean section rates cited in this paper from a referral facility, is important to report on the details associated with patient access, delays, and referral.

3. It is counterintuitive to believe that skills training would not accrue positive benefit, but the author correctly point out that the diagnosis of dystocia or fetal distress are often unclear and may lead to unnecessary operative deliveries.

4. Socio-economic status, distance from a hospital and whether a woman obtained prenatal care has been shown to be important in predicting risk.

From the data presented, it is difficult to prioritize future training needs. However, the importance of assuring generalizability among sites and reporting on antenatal risk factors would be helpful in analyzing future outcome data.

Reviewer #2: The author has modified most of the questions according to my revision opinions, except for a few minor questions, namely:Table5-table7 were not standardized.

Reviewer #3: 1. Please add the training time to the “Intervention” section, including when to start the training and the duration of the training.

2. There is still room for improvement in the description and interpretation of the results. For example, In “Abstract” section “after the implementation of health staff training, there were fewer incidence rate of dystocia, foetal distress and neonatal death, but more incidence rate of scarred uterus, post-operative complications and low birth weight.”

It is not appropriate to use obstetric characteristics of maternal (dystocia, foetal distress, scarred uterus) to evaluate the effect of intervention. This description makes readers feel that training is the reason for the increase of scarred uterus. It is more appropriate to evaluate the training effect with outcome indicators, however, low birth weight is not recommended as an evaluation indicator of intervention effect. It is meaningful to explore the influencing factors of low birth weight, but the purpose of this paper is to evaluate the effect of training.

3. In the paper, “Total duration of stay in days”， “patient’s file is complete”， “blood transfusions performed” are reported, and there are statistical significances between the two groups. These indicators are appropriate to assess the effect of training. So it is suggested to describe the relevant results in the abstract.

4. Although the results of data analysis do not suggest that training is effective in reducing neonatal death and post-operative complications, the research is meaningful. Analyzing the reasons behind the result and putting forward targeted countermeasures for risk factors of neonatal death and post-operative complications will increase the practical significance of this study.

7. PLOS authors have the option to publish the peer review history of their article (what does this mean?). If published, this will include your full peer review and any attached files.

Reviewer #1: No

Reviewer #2: No

Reviewer #3: No

---

## [Author Response · Author response to Decision Letter 1]

30 Nov 2021

Dear reviewers, I am very pleased to have had you for the evaluation of this study conducted in this region of the Democratic Republic of the Congo very difficult because of the war for several years. I will certainly take into account all your recommendations in our next study and I would like to take this opportunity to thank you for your precious time in helping us improve this work.I will be sure to recommend my other colleagues and researchers to your journal

---

## [Editor Report · Decision Letter 2]

13 Dec 2021

PONE-D-20-41094R2EVOLUTION IN CAESAREAN SECTION PRATICES IN NORTH-KIVU; IMPACT OF CAREGIVER TRAININGPLOS ONE

Dear Dr. dikete ekanga,

Thank you for submitting your manuscript to PLOS ONE. After careful consideration, we feel that it has merit but does not fully meet PLOS ONE’s publication criteria as it currently stands. Therefore, we invite you to submit a revised version of the manuscript that addresses the points raised during the review process. The authors did a nice job addressing comments and suggestions. However, I have a few minor additional comments.In order to provide a more complete information to our readers on the topic, we would like to emphasize the importance to cross referencing very recent material on the same topic published in "PLoS ONE ". Therefore, it would be highly appreciated if you would check the contents published in the last two years of "PLoS ONE" (https://journals.plos.org/plosone/) and add all material relevant to your article to the reference list.

We look forward to receiving your revised manuscript.

Kind regards,

Kamal Gholipour, PhD

Academic Editor

PLOS ONE
---

## [Editor Report · Decision Letter 3]

8 Feb 2022

EVOLUTION IN CAESAREAN SECTION PRACTICES IN NORTH-KIVU; IMPACT OF CAREGIVER TRAINING

PONE-D-20-41094R3

Dear Dr. dikete ekanga,

We’re pleased to inform you that your manuscript has been judged scientifically suitable for publication and will be formally accepted for publication once it meets all outstanding technical requirements.

Kind regards,

Kamal Gholipour, PhD

Academic Editor

PLOS ONE
---

## [Editor Report · Acceptance letter]

17 May 2022

PONE-D-20-41094R3 

EVOLUTION IN CAESAREAN SECTION PRACTICES IN NORTH KIVU: IMPACT OF CAREGIVER TRAINING 

Dear Dr. Dikete Ekanga:

I'm pleased to inform you that your manuscript has been deemed suitable for publication in PLOS ONE. Congratulations! Your manuscript is now with our production department. 

Kind regards, 

on behalf of

Dr. Kamal Gholipour 

Academic Editor

PLOS ONE